# Associations of Microbial Diversity with Age and Other Clinical Variables among Pediatric Chronic Rhinosinusitis (CRS) Patients

**DOI:** 10.3390/microorganisms11020422

**Published:** 2023-02-07

**Authors:** Shen Jean Lim, Warit Jithpratuck, Kathleen Wasylik, Panida Sriaroon, Larry J. Dishaw

**Affiliations:** 1Department of Pediatrics, Morsani College of Medicine, University of South Florida, Tampa, FL 33620, USA; 2Pediatric Ear, Nose & Throat Specialists, Johns Hopkins All Children’s Hospital, St. Petersburg, FL 33701, USA; 3USF Pediatric Allergy/Immunology Clinic, Food Allergy Clinic, Johns Hopkins All Children’s Hospital, St. Petersburg, FL 33701, USA

**Keywords:** chronic rhinosinusitis, microbiome, adenoid, pediatrics

## Abstract

Chronic rhinosinusitis (CRS) is a heterogenous disease that causes persistent paranasal sinus inflammation in children. Microorganisms are thought to contribute to the etiology and progression of CRS. Culture-independent microbiome analysis offers deeper insights into sinonasal microbial diversity and microbe–disease associations than culture-based methods. To date, CRS-related microbiome studies have mostly focused on the adult population, and only one study has characterized the pediatric CRS microbiome. In this study, we analyzed the bacterial diversity of adenoid tissue, adenoid swab, maxillary sinus, and sinus wash samples from 45 pediatric CRS patients recruited from the Johns Hopkins All Children’s Hospital (JHACH) in St. Petersburg, FL, USA. The alpha diversity in these samples was associated with baseline nasal steroid use, leukotriene receptor antagonist (LTRA) use, and total serum immunoglobulin (Ig) E (IgE) level. *Streptococcus*, *Moraxella*, and *Haemophilus* spp. were most frequently identified from sinus cultures and the sequenced 16S rRNA gene content. Comparative analyses combining our samples with the samples from the previous microbiome study revealed differentially abundant genera between patients with pediatric CRS and healthy controls, including *Cutibacterium* and *Moraxella*. Additionally, the abundances of *Streptobacillus* and *Staphylococcus* were consistently correlated with age in both adenoid- and sinus-derived samples. Our study uncovers new associations of alpha diversity with clinical parameters, as well as associations of specific genera with disease status and age, that can be further investigated.

## 1. Introduction

Pediatric chronic rhinosinusitis (CRS) is a persistent and heterogenous inflammatory disease of paranasal sinuses, with a prevalence of around 5–15% in children [1,2,3]. CRS is generally defined by the presence of two or more symptoms of facial pain/pressure, nasal congestion, nasal drainage, cough, hyposmia/anosmia, or cough over three months [4,5] and confirmed by nasal endoscopy and/or CT scan [6]. First-line, non-surgical interventions for pediatric CRS include nasal saline spray, nasal steroid spray, and oral antibiotics [6]. In pediatric CRS, the adenoids are often enlarged and harbor microbial biofilms, blocking the drainage of nasal discharge [3]. As such, adenoidectomy, combined with maxillary sinus antral washout, may be considered if non-surgical options fail [3,6]. Functional endoscopic sinus surgery (FESS) is a treatment option for pediatric CRS patients who failed adenoidectomy and/or non-surgical treatments, or if sinus blockages are observed or comorbidities are present [3,6,7].

To date, the etiology of pediatric CRS is not well understood. The condition is thought to be caused by a diverse range of intrinsic (immune- or anatomy-related) factors and extrinsic (environment- or microbe-related) factors [8,9]. Among the extrinsic factors, there is a growing research interest in the potential contributions of bacterial communities to the etiology and progression of pediatric CRS. Culture-based studies identified a combination of aerobic and anaerobic bacteria from pediatric and adult CRS patients [4]. Bacteria cultured from pediatric patients included *Bacteroides*, *Prevotella*, and *Prevotella melaninogenica* (previously *Bacteroides melaninogenicus*) from class Bacteroidota; *Streptococcus* (alpha-hemolytic *Streptococcus*, group A *Streptococcus*, and *Streptococcus pneumoniae*); *Staphylococcus* (coagulase-negative *Staphylococcus* and *Staphylococcus aureus*); *Peptostreptococcus* from class Firmicutes, *Fusobacterium* and other members of class Fusobacteriota; and *Haemophilis influenzae* and *Moraxella catarrhalis* from class Gammaproteobacteria [10,11]. However, the frequencies of these bacterial isolates in pediatric CRS patients were inconsistent between studies. Besides culture-based studies, microscopy-based studies have detected bacterial biofilms covering >80% of the adenoidal surfaces in pediatric CRS patients [12,13]. Another study incorporating both microscopy and 16S rRNA polymerase chain reaction (PCR) amplification showed that 54% of the 39 adenoid specimens studied had biofilms, which may contain *Corynebacterium argentoratense*, *Micrococcus luteus*, *Staphylococcus aureus*, and *Streptococcus salivarius* [14].

More recently, culture-independent sequencing-based approaches targeting the 16S rRNA phylogenetic marker gene have provided more information on the microbial diversity of the upper respiratory tract, which includes culturable and unculturable microorganisms [15]. Previous studies analyzed the concordance between culturing and sequencing results [16,17,18,19], evaluated specimen types for CRS-related microbiome studies [17,19], and investigated differences in microbial diversity between patients and healthy controls, often in association with clinical parameters [18,19,20,21,22,23]. Results from these studies were, however, inconsistent, likely due to differences in study design [17]. Furthermore, almost all microbiome studies focus on the adult population [15] and there is a paucity of microbiome studies focusing on pediatric CRS, partly because antibiotic therapy perturbs the microbial communities of the patients [10]. To our knowledge, the only study comparing the microbiomes of pediatric CRS patients (*n* = 37) and healthy controls (*n* = 50) was performed by Stapleton et al. at the University of Pittsburgh Medical Center (UPMC) [23]. The study identified *Moraxella* as the most abundant bacterial taxon across all samples and revealed microbial alpha and beta diversity differences between nasopharynx (adenoid bed) and nasal cavity (sinus) samples [23]. Alpha diversity in the samples was associated with clinical variables, including age, eczema, nasal steroid use, inhaled steroid use, and antihistamine use [23]. However, the study did not find any differences in microbial diversity between pediatric patients and healthy controls [23].

To further understand adenoid and sinus microbial diversity in pediatric CRS patients, we analyzed the bacterial 16S rRNA gene content of samples collected from 45 children recruited from the Johns Hopkins All Children’s Hospital (JHACH) in St Petersburg, FL, USA. We compared bacterial diversity across adenoid tissue, adenoid swab, maxillary sinus, and sinus wash samples in association with culturing results and clinical metadata. Because our experimental design did not permit sample collection from healthy controls, we also included Stapleton et al.’s healthy control and patient data from the UPMC [23] in our analysis pipeline for detailed microbiome comparisons across disease status and across cohorts.

## 2. Materials and Methods

### 2.1. Enrollment and Sample Collection

Following approval from the Institutional Review Board (JH-IRB00039391), participants were recruited at JHACH from 2015 to 2018. The original inclusion criteria were broad and included subjects between 9 months and 55 years of age who already had plans to undergo rhinoscopy or FESS as clinically indicated for treatment of CRS or non-CRS-related complaints; were willing to provide informed consent and assent when applicable; and were willing to follow the study’s schedule. At least 50% of the subjects must be under the age of 21. The broader study protocol targeted four study groups to evaluate contributions from immune dysfunction if diagnosis was known: common variable immune deficiency (CVID) with CRS; selective IgA deficiency (SIgAD) with CRS; CRS without immune deficiency; and those without CRS. Where possible, diagnosis was assisted by CT scan performed as part of standard care. Exclusion criteria included any other immune deficiency, use of immunomodulating or immunosuppressive medications, and any condition that, in the opinion of the investigator, would interfere with the conduct of the study. Because appropriate response to vaccination is used to define CVID, patients with no history of vaccination were excluded. Pregnant patients were also excluded.

In the current reported study, 45 pediatric CRS participants were enrolled. Adenoid tissue samples were collected from 40 participants undergoing adenoidectomy (Table 1). Duplicate adenoid swab samples from 14 of these participants were also collected; one sample was sent for routine bacterial culture while the other was sent for sequencing. A portion of the sinus biopsy samples from 15 of these participants was also sent for routine bacterial culture (Table 1). Biopsy and wash samples were collected from the left and/or right maxillary sinuses of five other pediatric participants undergoing FESS and used for microscopy, bacterial culture, and sequencing (Table 1). Microscopy procedures included Gram-staining, acid-fast staining, and/or fungal smear, while culturing procedures were used to detect respiratory microbes, anaerobic microbes, and/or fungi. Additionally, serum samples were collected by venipuncture from seven participants undergoing adenoidectomy and five participants undergoing FESS to measure serum IgE levels. Among participants with measured IgE levels, serum samples from a subset of two participants undergoing adenoidectomy and four participants undergoing FESS were also tested for IgG, IgA, and IgM levels (Table 1). Demographic and clinical metadata, including sex, age, atopic conditions, and medication history, were collected from the medical records (Table 1).

Samples for sequencing were preserved in RNAlater and stored at −80 °C. DNA was extracted from the samples using the Qiagen PowerSoil Kit. The V4 region of the 16S rRNA gene was amplified using the 515F/806R primers and sequenced by Diversigen^®^ (previously known as CoreBiome; Saint Paul, MN, USA).

### 2.2. Data Analysis

Raw reads and metadata from the UPMC cohort were retrieved from the National Center for Biotechnology Information Sequence Read Archive (NCBI SRA) using the BioProject accession PRJNA634373. These were combined with sequenced reads and metadata from the JHACH cohort in this study and imported into QIIME2 v2020.6 [24]. All reads were denoised by DADA2 [25] into amplicon sequence variants (ASVs) and clustered *de novo* into operational taxonomic units (OTUs) at 99% identity to reduce batch effects [26]. A phylogenetic tree was generated from the OTU sequences using the SATé-Enabled Phylogenetic Placement (SEPP) fragment insertion method [27]. Each OTU was taxonomically classified using a naïve Bayes classifier trained on the 515F/806R V4 region of 16S rRNA gene sequences from the SILVA v138 database [28]. Following classification, only OTUs with phylum-level annotations were retained, while OTUs annotated as archaea, chloroplast, eukaryote, and mitochondria were removed from the count table. The filtered count table was rarefied to 1003 OTUs per sample (the smallest 4-digit number), resulting in the elimination of 22 samples. These included JHACH samples from three sinus washes, one sinus tissue biopsy, three adenoid tissues, and one adenoid swab (Appendix A), as well as four UPMC’s reagent blanks from New England Biolabs^®^ (Ipswich, MA, USA) Q5 HS High-Fidelity polymerase and 10 negative controls from the Qiagen (Germantown, MD, USA) DNeasy Powersoil Kit.

The normality of clinical variables and alpha diversity variables was evaluated using Shapiro-Wilk tests implemented in R v4.0.2 [29], which rejected the null hypothesis that these variables were normally distributed. Ages between subgroups were compared using non-parametric Mann–Whitney-U tests which accept the numerical variable, while gender between subgroups was compared using the non-parametric Kruskal–Wallis test which accepts the categorical variable. Alpha diversity was computed from the rarefied count table using QIIME2 [24]. Alpha diversity variables between subgroups were compared using pairwise Mann–Whitney-U tests and the *p*-values were adjusted for multiple comparisons using the false discovery rate. Correlation analyses were performed separately on each sample subgroup using non-parametric Spearman’s rank correlation.

For beta diversity, principal components analysis (PCA) plots were computed using the ampvis2 v2.6.5 [30] R package from rarefied and Hellinger-transformed [31] OTU counts with ≥0.1% relative abundance. The envfit function in vegan v2.5.7 [32] was used to fit categorical or numerical clinical variables onto the PCA plot. The most abundant genera in each sample group were identified using ampvis2 [30] from the rarefied OTU count table. Differentially abundant genera across cohort- and tissue-type subgroups were predicted using the analysis of composition of microbiomes (ANCOM) [33] plugin in QIIME2 [24] from the unrarefied count table. Prior to ANCOM analysis, the unrarefied count table was collapsed at the genus level and imputed pseudocounts were added to generate non-zero genera abundance values in QIIME2. Within each tissue type (adenoid tissue, adenoid swab, sinus tissue biopsy, sinus wash) collected from the JHACH cohort, the MaAsLin2 v1.4.2 R package [34] was used to identify statistical associations of unrarefied genus-relative abundances with all clinical metadata, using the default parameters of total sum scaling (TSS) normalization, log-transformation, linear model (LM), and Benjamini–Hochberg (BH) [35] correction for multiple testing. Prior to the MaAsLin2 analyses, the collapsed genus count table was converted into a relative abundance table using QIIME2.

## 3. Results

### 3.1. Participant Characteristics

The JHACH cohort was part of a larger study (IRB 00039391) and comprised 40 participants undergoing adenoidectomy and five participants undergoing FESS as treatment modalities for CRS (Table 1). Adenoidectomy patients were 1–17 years old and 65% male (*n* = 26; Table 1). From these patients, 41 adenoid tissue samples and 13 adenoid swab samples were sequenced (Table 1). FESS patients were 8–17 years old and 40% male (*n* = 2; Table 1). From these patients, eight sinus biopsy samples and nine sinus wash samples were sequenced (Appendix A). Total serum IgE levels from seven participants undergoing adenoidectomy and five participants undergoing FESS were all above the reference of 25 IU/mL (Table 1). Among participants with measured IgE levels, serum IgG, IgA, and IgM levels were also tested for a subset of two participants undergoing adenoidectomy and four participants undergoing FESS (Table 1). Measured IgG, IgA, and IgM levels were within the age-matched reference range [36].

### 3.2. Sinus Culture Results

*Streptococcus pneumoniae* (*n* = 6), *Moraxella catarrhalis* (*n* = 6), and *Haemophilus influenza* (*n* = 6) were the most common bacterial species identified in the sinus cultures of 15 of the JHACH participants undergoing adenoidectomy (Table 2). Lower occurrences of *Staphylococcus aureus* (*n* = 2), *Streptococcus pyogenes* (*n* = 1), *Pseudomonas aeruginosa* (*n* = 1), and Gram-positive cocci (*n* = 1) were observed in this group of participants. The sinus culture from one JHACH participant undergoing FESS grew *Streptococcus pneumoniae* while the culture from another participant grew *Curvularia* fungal species.

### 3.3. Alpha Diversity

Amplicon-sequenced reads from the UPMC [23] and JHACH cohorts were combined and analyzed using the QIIME2 pipeline. There were no significant differences in age (Mann–Whitney-U *p* = 0.4) and gender (Kruskal–Wallis *p* = 0.3) between participants from the JHACH and UPMC cohorts. Demographics of the study population in the UPMC cohort are presented in Table 1 of Stapleton et al. [23].

The 16S rRNA gene analysis of the raw reads from the JHACH and UPMC cohorts produced 2502 bacterial OTUs clustered at 99% identity. After rarefying to 1003 OTUs per sample, 1152 OTUs remained in the rarefied count table. Most PCR and extraction kit negative controls sequenced from the UPMC cohort contained <1003 OTUs per sample, including four out of five (80%) PCR reagent blanks and 10 out of 14 (71%) extraction kit negative controls. The number of OTUs in these negative controls was significantly lower compared to other sinonasal tissue samples (Figure 1 and Appendix A).

Alpha diversity, which represents the richness (number of bacterial OTUs) and/or the evenness (abundance distribution of the bacterial OTUs) within each sample, was calculated for each sample using Faith’s phylogenetic diversity, observed OTUs, Shannon diversity, and Pielou’s evenness metrics [26,37]. Adenoid samples from JHACH adenoidectomy patients generally showed the highest alpha diversity across subgroups (adjusted *p*-value of 0.05; Figure 1 and Appendix A). No significant differences in alpha diversity were observed between the JHACH adenoid samples and adenoid swab samples from the UPMC healthy controls (Figure 1 and Appendix A). The JHACH adenoidectomy patients with a history of pre-operative nasal steroid use showed higher alpha diversity in their adenoid samples than those with no documented history of nasal steroid use (Figure 2a). The same trend was observed in the JHACH adenoidectomy patients with a history of LTRA use (Figure 2b). Additionally, Faith’s phylogenetic diversity was significantly correlated with serum IgE levels (*p* = 0.03), but this trend was not observed for the other alpha diversity metrics.

Adenoid swabs had lower alpha diversity than adenoid samples in the JHACH cohort (Figure 1 and Appendix A). Adenoid swabs from JHACH adenoidectomy patients also showed lower alpha diversity than adenoid swabs from the UPMC and healthy participants (Figure 1 and Appendix A). Because of sample size limitations, alpha diversity can only be compared between JHACH adenoid swab samples from patients with and without a history of antihistamine use. These comparisons were not statistically significant.

Sinus wash samples from JHACH FESS patients did not significantly differ in alpha diversity from sinus biopsy samples from JHACH FESS patients (Figure 1 and Appendix A). However, the Shannon index and Pielou’s evenness of sinus wash samples from JHAC FESS patients were higher compared to those predicted in sinus swab samples from CRS patients and healthy patients of the UPMC cohort (Figure 1 and Appendix A). Because of the small sample sizes, alpha diversity in sinus wash and sinus biopsy samples from the JHACH cohort can only be analyzed with IgE, IgG, IgA, and IgM levels and history of antibiotic use (sinus biopsy samples only). No signification associations were observed between the alpha diversity values and the clinical metadata.

Adenoid and sinus swabs from all subgroups of the UPMC cohort, except for sinus swabs from CRS patients, showed statistically significant correlations between alpha diversity and age (Appendix A). These age-related correlations were not observed in the JHACH cohort (Appendix A).

### 3.4. Beta Diversity

Beta diversity analyses were performed to analyze dissimilarities in bacterial composition across sample subgroups [26]. Microbial composition across all samples was significantly influenced by age (*p* = 0.001) and cohort/treatment/sample type subgroups (*p* = 0.001; Figure 3a). The significant association of age with beta diversity ordination scores was consistently observed in sample subsets containing only JHACH CRS patients (*p* = 0.001; Figure 3b), UPMC CRS patients (*p* = 0.027; Figure 3c), and UPMC healthy controls (*p* = 0.001; Figure 3d).

Spearman’s correlation analyses revealed 37 and 31 unique genera whose abundances were significantly correlated with age in the adenoid (Appendix A) and sinus (Appendix A) samples from both cohorts, respectively. *Streptobacillus* abundances were consistently negatively correlated with age in all adenoid sample subgroups (Figure 4a), while *Staphylococcus* abundances were consistently positively correlated with age in most sinus sample subgroups, except for JHACH sinus wash samples where no age-correlated genus was identified (Figure 4b).

### 3.5. Taxonomic Composition in Adenoid-Derived Samples

Consistent with the culture results, *Streptococcus*, *Haemophilus*, and *Moraxella* were the most abundant genera in all adenoid-derived samples (Figure 5). A total of 16 differentially abundant genera were identified between adenoid-derived samples from different cohorts and subgroups. Most differentially abundant genera (*n* = 13) were among the 30 most abundant genera in the adenoid-derived samples (Figure 5), except for *Actinomyces*, *Cutibacterium*, and *Yersiniaceae* (unassigned genus). In the UPMC cohort, no differentially abundant genera were identified between adenoid swab samples from CRS patients and healthy controls, as previously reported [23].

Compared to adenoid swab samples from the UPMC healthy controls, adenoid swab samples from JHACH adenoidectomy patients had higher abundances of *Burkholderia*-*Caballeronia*-*Paraburkholderia*, *Cutibacterium*, and *Yersiniaceae* (unassigned genus), but lower abundances of *Actinomyces*, *Alloprevotella*, *Campylobacter*, *Fusobacterium*, *Gemella*, *Granulicatella*, *Leptotrichia*, *Neisseria*, *Porphyromonas*, *Prevotella*, *Rothia*, and *Veillonella* (Figure 5). Among adenoid swab samples from CRS patients, those from the JHACH cohort showed higher abundances of *Burkholderia*-*Caballeronia*-*Paraburkholderia* and *Haemophilus* and lower abundances of *Fusobacterium* compared to those from the UPMC cohort (Figure 5). Within the JHACH cohort, adenoid samples were significantly enriched in *Alloprevotella*, *Prevotella*, and *Veillonella* compared to adenoid swab samples (Figure 5). MaAsLin2 [34] did not detect any significant associations between genera-relative abundances and the clinical metadata in adenoid and adenoid swab samples within the JHACH cohort.

### 3.6. Taxonomic Composition in Sinus-Derived Samples

Like the adenoid-derived samples, *Moraxella*, *Haemophilus*, and *Streptococcus* were the most abundant genera in all sinus-derived samples (Figure 6). However, their abundances were uneven between sample subgroups. As with the adenoid swab samples and as previously reported [23], in the UPMC cohort, no differentially abundant genera were identified between sinus swab samples from CRS patients and healthy controls.

Compared to sinus swab samples from the UPMC healthy controls, both sinus biopsy and sinus wash samples from JHACH patients undergoing FESS had higher *Cutibacterium* abundances and lower *Moraxella* abundances (Figure 6). The JHACH sinus biopsy samples, but not sinus wash samples, were enriched in *Escherichia*-*Shigella* (Figure 6) and *Yersiniaceae* (unassigned genus) compared to sinus swabs from UPMC healthy controls. Additionally, JHACH sinus wash samples, but not biopsy samples, had higher *Pasteurella* abundances and lower *Dolosigranulum* abundances relative to sinus swabs from the UPMC healthy controls (Figure 6). All differentially abundant genera identified were among the 35 most abundant genera in all sinus-derived samples (Figure 6).

Within the JHACH cohort, 121 genera from the phyla Actinobacteriota (*n* = 20), Bacteroidota (*n* = 13), Campilobacterota (*n* = 1; *Campylobacter*), Deinococcota (*n* = 2; *Deinococcus* and *Thermus*), Firmicutes (*n* = 50), Fusobacteriota (*n* = 3; *Leptotrichia*, *Fusobacterium*, and *Streptobacillus*), Gemmatimonadota (*n* = 1; *Gemmatimonas*), Patescibacteria (*n* = 2; Saccharimonadales and Saccharimonadaceae TM7x), and Proteobacteria (*n* = 29) were differentially abundant when drawing comparisons between sinus wash and sinus swab samples (Appendix A). Genera-relative abundances did not show any significant associations with clinical metadata among the JHACH sinus biopsy samples and JHACH sinus wash samples.

## 4. Discussion

Pediatric CRS is a heterogenous and poorly understood disease that can be influenced by various factors associated with the immune system, anatomy, environment, and sinonasal bacterial communities [8,9]. To investigate potential associations of the sinonasal microbiome with pediatric CRS, we analyzed the bacterial 16S rRNA gene content of adenoid tissue, adenoid swab, maxillary sinus, and sinus wash samples from pediatric CRS patients recruited from JHACH. Bacterial diversity within the JHACH cohort was analyzed in association with culture results and clinical metadata and in association with previously sequenced samples from healthy participants and pediatric CRS patients recruited from the UPMC [23].

Culture results [23] and microbial diversity identified from the JHACH cohort samples were consistent with previous reports. *Streptococcus pneumoniae*, *Moraxella catarrhalis*, and *Haemophilus influenza* were the most frequent species observed from the adenoid cultures, while sinus cultures obtained from two JHACH participants undergoing FESS contained *Streptococcus pneumoniae* and a *Curvularia* fungal species, although sample sizes in our study were small. These bacterial species are commonly cultured from pediatric CRS patients [10,11]. In agreement with the culture results, the genera *Streptococcus*, *Moraxella*, and *Haemophilus* were the most abundantly sequenced in the adenoid-derived and sinus-derived samples from the JHACH cohort. *Moraxella* was also the most abundant bacterial taxon among adenoid swab and sinus swab samples sequenced from the UPMC cohort [23]. Although fungal sinonasal infections are rare in children [38], *Curvularia* is one of the most common fungal pathogens cultured from patients with allergic fungal sinusitis [39]. Future meta-omics approaches characterizing the taxonomic and functional abundances of bacteria, archaea, fungi, and viruses will be useful in providing a global overview of microbial changes associated with pediatric CRS.

Within the adenoid samples of the JHACH cohort, we observed significant associations of alpha diversity with pre-operative nasal steroid use, LTRA use, and potentially total serum IgE levels. The previous UPMC cohort study predicted higher alpha diversity in the nasopharynx compared to the nasal cavity, and inhaled steroid use increased the Shannon diversity differences between both locations [23]. The UPMC study also identified negative associations between LTRA use and *Prevotella* abundances in the nasopharynx relative and nasal cavity [23]. A longitudinal pilot study on four adult subjects with CRS revealed shifts in microbial beta diversity before and after administration of the nasal steroid mometasone furoate, such as an increase in *Staphylococcus* relative abundance and a decrease in *Moraxella* and *Streptococcus* abundances [40]. However, in the JHACH cohort, genera abundances were not significantly associated with drug use. The effects of nasal steroids and LTRA—which are widely used to treat CRS and asthma—on sinonasal microbial diversity are poorly characterized and require further study. Given the anti-inflammatory effects of nasal steroids and LTRA [41], we speculate that the use of these drugs would increase alpha diversity as inflammation levels decrease, as previously observed in the gut microbiome [42]. Similarly, the utility of total IgE serum levels in predicting alpha microbial diversity and CRS disease status remains to be evaluated. A previous study showed significant correlations between total IgE serum levels with the Sino-Nasal Outcome Test scores, Lund–Mackay computed tomography scan scores, and polyposis recurrence in 50 adult CRS patients with nasal polyposis, along with interleukin-17 and pentraxin-3 levels [43]. Nevertheless, there is a paucity of studies on the associations between clinical variables, such as drug use and total IgE serum levels, and microbial diversity with CRS disease progression.

Significant correlations between age and microbial beta diversity were consistently observed in all samples, as well as in the subgroups of samples from JHACH CRS patients, UPMC CRS patients, and UPMC healthy controls. We identified two genera whose abundances were significantly and consistently correlated with age in adenoid- and sinus-derived samples. *Streptobacillus*, previously reported to be enriched in adenoid samples of Korean children with otitis media with effusion [44], were negatively correlated with age in the adenoid-derived samples, while *Staphylococcus* abundances were positively correlated with age in most sinus sample subgroups. Differences in bacterial community composition were also documented between younger CRS patients (21 to 59 years old) and older CRS patients (≥60 years old) [45], and similar age-related changes in *Staphylococcus aureus* population structures were observed in younger and older cystic fibrosis patients [46]. In young individuals <21 years old, age was similarly associated with nasopharyngeal microbial diversity and found to influence associations between several bacterial taxa, SARS-CoV-2 infection status, and SARS-CoV-2 symptoms [47]. In our study, the correlations between *Streptobacillus* and *Staphylococcus* abundances with age were observed both in CRS patients and healthy controls. As 16S rRNA gene sequencing only provides limited resolution at the genus level, species- or strain-level meta-omics analysis will be useful in tracking sinonasal microbiome maturation and identifying age-dependent microbial markers associated with CRS progression and outcomes at a higher resolution. Age was also correlated with alpha diversity in the UPMC cohort, but not the JHACH cohort. The difference in alpha diversity results between cohorts could be due to differences in sample sizes and sampling strategies.

Our study expands on the previous UPMC study on the CRS microbiome [23] by providing additional sample types, particularly adenoid tissue samples and sinus wash samples, for comparisons. The alpha diversity of adenoid samples from JHACH adenoidectomy patients was higher than most subgroups of sinus swab, sinus wash, and adenoid swab samples, except for adenoid swab samples from UPMC healthy controls. This agrees with a previous study which showed high alpha diversity in adenoid samples and suggested that adenoid samples may be a more reliable sampling site for microbial diversity due to lower contamination from nasal cavity, nasopharynx, or oral cavity [44]. In the JHACH cohort, although the alpha diversity in the sinus wash samples and sinus biopsy samples was not significantly different, 121 genera were differentially abundant between the sinus wash and sinus biopsy samples. Additionally, JHACH sinus wash samples had higher Shannon index and Pielou’s evenness compared to sinus swab samples from CRS patients and healthy patients in the UPMC cohort. This suggests that sinus wash samples and sinus biopsy samples differ in microbial diversity. While the utility of sinus wash samples in the CRS microbiome studies remains to be explored, our results highlight the need for consistent and representative sampling sites and strategies to evaluate microbe–disease associations more accurately.

Because our study did not recruit healthy controls, we included microbiome data from the UPMC healthy controls, as well as UPMC CRS patients, for comparison between cohorts. Our results confirmed previously reported results where no differentially abundant genera were identified from the adenoid and sinus swabs between healthy and CRS patients from the UPMC cohort [23]. However, comparisons between adenoid-derived samples sequenced from JHACH CRS patients and UPMC healthy controls revealed several differentially abundant genera. *Burkholderia*-*Caballeronia*-*Paraburkholderia*, *Cutibacterium*, and *Yersiniaceae* (unassigned genus) were enriched in JHACH CRS patients, while *Actinomyces*, *Alloprevotella*, *Campylobacter*, *Fusobacterium*, *Gemella*, *Granulicatella*, *Leptotrichia*, *Neisseria*, *Porphyromonas*, *Prevotella*, *Rothia*, and *Veillonella* were enriched in UPMC healthy controls. We also consistently observed higher *Cutibacterium* abundances and lower *Moraxella* abundances in sinus-derived samples from JHACH patients undergoing FESS compared to UPMC healthy controls. *Burkholderia*-*Caballeronia*-*Paraburkholderia* [48,49] and *Yersiniaceae* [50] are not typically associated with CRS and their associations with the disease require further investigation. *Cutibacterium* species, including *C. acnes*, *C. granulosum*, and *C. avidum*, were the most common anaerobes identified by mass spectrometry in sinus samples from CRS patients [51,52]. Despite the high overall *Moraxella* abundances across all samples, its relatively lower frequency in sinus tissue biopsy and sinus swab samples from JHACH CRS patients compared to UPMC healthy controls is surprising, given that *Moraxella* is frequently identified in the pediatric CRS population [10,23,38]. The lower *Moraxella* abundances could be attributed to antibiotic use [10] or nasal steroid use [40]. However, MaAsLin2 [34] did not report significant associations between genera-relative abundances and clinical variables, including drug use, in sinus-derived samples from the JHACH cohort. This could be due to the small sample size, since sinus-derived samples were only collected from five JHACH CRS patients undergoing FESS. Future large-scale studies with sufficient replicates from distinct subgroups of CRS patients and healthy controls will facilitate in-depth analyses on the roles and interactions of various clinical and microbial factors on pediatric CRS.

## 5. Conclusions

Our study confirms the association between microbial diversity and age in pediatric CRS patients and presents new findings on differentially abundant taxa between pediatric CRS patients and healthy controls. With small sample sizes, our study is underpowered to detect subtle differences in microbial communities in each sub-stratified group. Since pediatric CRS is a heterogenous disease with high inter-subject microbiota variability [9], future microbiome studies with larger cohorts will be helpful in validating these CRS-associated microbial taxa and in analyzing the associations between clinical variables and sinonasal microbial diversity with higher resolution.

## Figures and Tables

**Figure 1 microorganisms-11-00422-f001:**
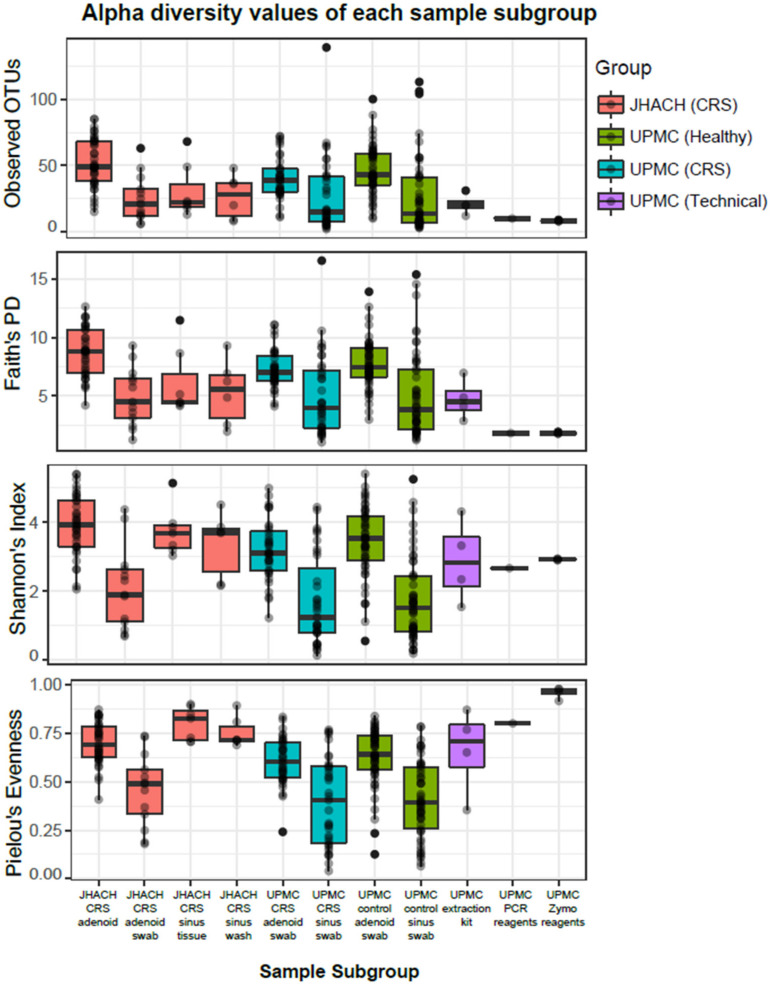
Alpha diversity values of each sample subgroup comprising samples from chronic rhinosinusitis (CRS) patients, healthy controls, or technical controls collected from the Johns Hopkins All Children’s Hospital (JHACH) or University of Pittsburgh Medical Center (UPMC) cohorts. Each point represents a sample and its corresponding alpha diversity value is represented on the *y*-axis. A high alpha diversity value indicates high bacterial richness and/or evenness within the sample. Samples were aggregated into subgroups corresponding to the cohort, disease status, and sample type, as indicated in the *x*-axis.

**Figure 2 microorganisms-11-00422-f002:**
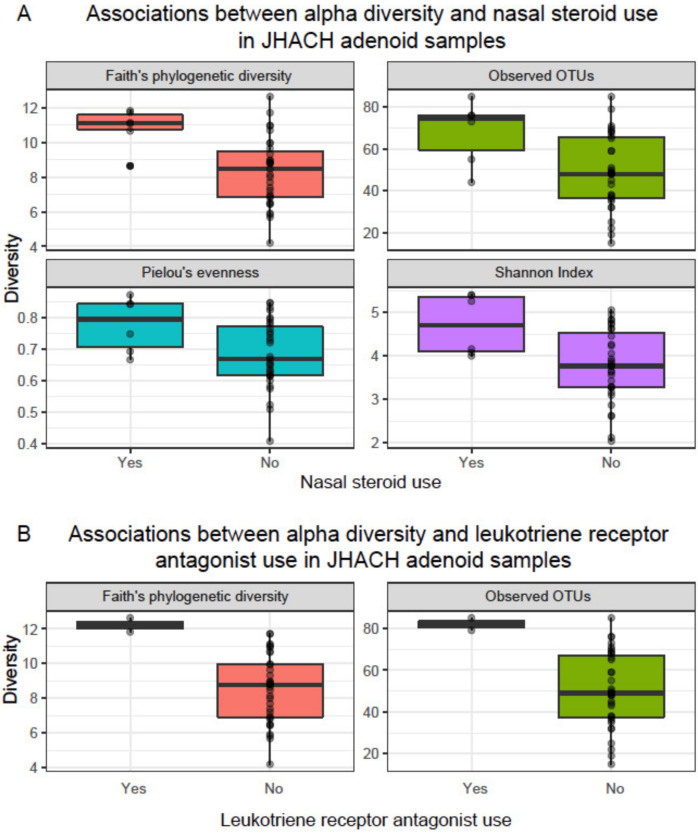
Statistically significant associations (*p* < 0.05) between alpha diversity and (**A**) nasal steroid use; and (**B**) leukotriene receptor antagonist (LTRA) use in adenoid samples from patients undergoing adenoidectomy in the Johns Hopkins All Children’s Hospital (JHACH) cohort. Alpha diversity was computed using Faith’s phylogenetic diversity, observed operational taxonomic units (OTUs), Pielou’s evenness, and the Shannon index.

**Figure 3 microorganisms-11-00422-f003:**
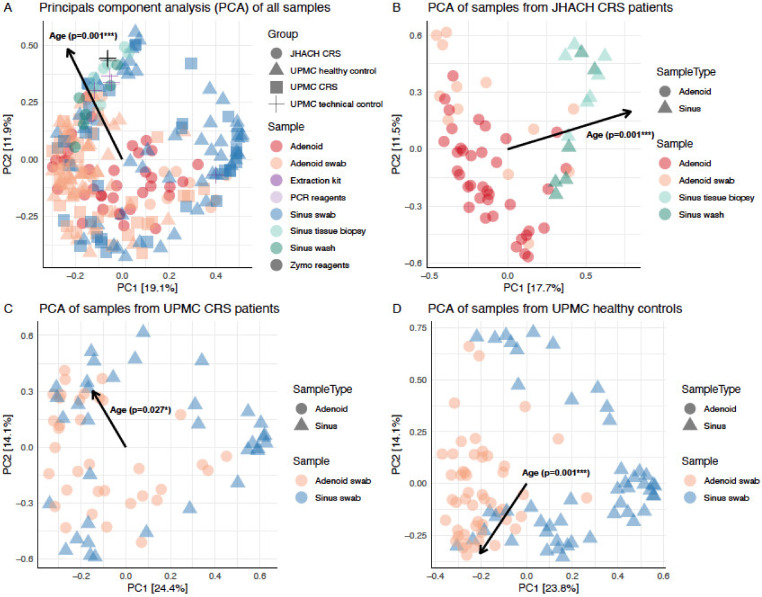
Principal components analysis (PCA) biplot showing the significant effects of age on the clustering of (**A**) all analyzed samples; (**B**) samples collected from chronic rhinosinusitis (CRS) patients in the Johns Hopkins All Children’s Hospital (JHACH) cohort; (**C**) samples collected from CRS patients in the University of Pittsburgh Medical Center (UPMC) cohort; and (**D**) samples collected from healthy controls in the UPMC cohort. Each point in the PCA biplot represents a sample. The percentages on each axis represent the relative contribution (eigenvalue) of each axis to the total inertia (variation) in the dataset. * denotes a *p*-value of ≤0.05 and *** denotes a *p*-value of ≤0.001.

**Figure 4 microorganisms-11-00422-f004:**
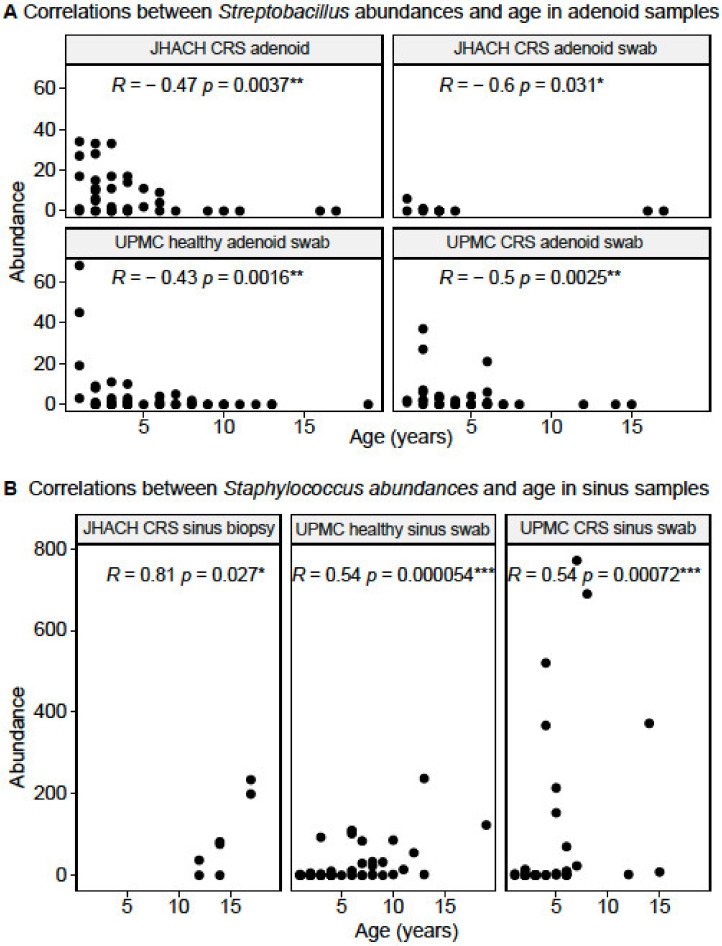
Spearman correlations between age and (**A**) *Streptobacillus*; and (**B**) *Staphylococcus* abundances in subgroups of adenoid- and sinus-derived samples, respectively, from chronic rhinosinusitis (CRS) and healthy participants in the Johns Hopkins All Children’s Hospital (JHACH) and/or University of Pittsburgh Medical Center (UPMC) cohorts. All correlations were statistically significant, where * denotes a *p*-value of ≤0.05; ** denotes a *p*-value of ≤0.01; and *** denotes a *p*-value of ≤0.001.

**Figure 5 microorganisms-11-00422-f005:**
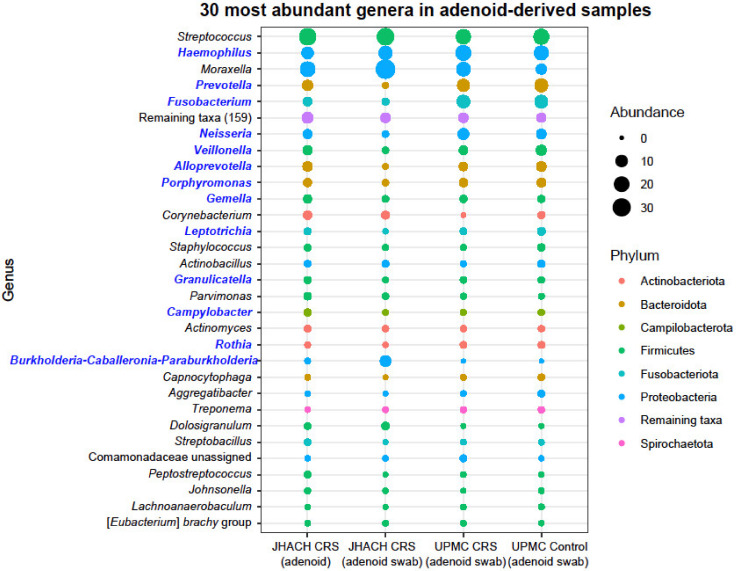
Mean abundances of the 30 most abundant genera in adenoid-derived samples from chronic rhinosinusitis (CRS) and healthy participants in the Johns Hopkins All Children’s Hospital (JHACH) and/or University of Pittsburgh Medical Center (UPMC) cohorts. Differentially abundant genera across sample subgroups are indicated in blue.

**Figure 6 microorganisms-11-00422-f006:**
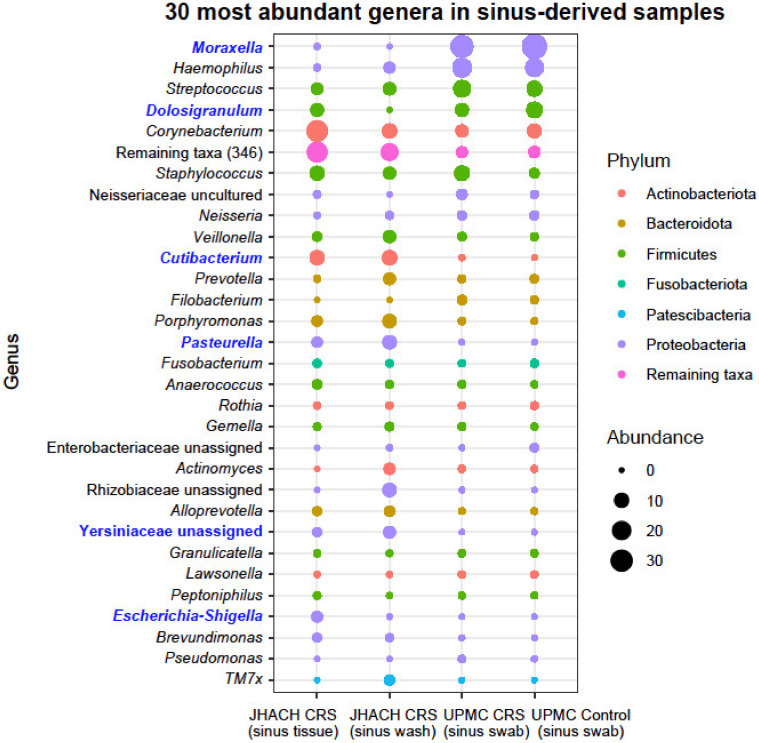
Mean abundances of the 30 most abundant genera in the sinus-derived sample from chronic rhinosinusitis (CRS) and healthy participants in the Johns Hopkins All Children’s Hospital (JHACH) and/or University of Pittsburgh Medical Center (UPMC) cohorts. Differentially abundant genera across sample subgroups are indicated in blue.

**Table 1 microorganisms-11-00422-t001:** Demographics of the study population in the Johns Hopkins All Children’s Hospital (JHACH) cohort. The cohort includes patients undergoing adenoidectomy and functional endoscopic sinus surgery (FESS). For numeric variables, the mean and standard deviation (SD) were calculated.

Variable	Adenoidectomy Patients *n* = 40	FESS Patients *n* = 5
Sex		
Female, *n* (%)	14 (35)	3 (60)
Male, *n* (%)	26 (65)	2 (40)
Age, mean ± SD (years)	4.3 ± 3.7	13 ± 3.3
Sinus culture		
Yes, *n* (%)	15 (37.5)	5 (100)
No, *n* (%)	25 (62.5)	0 (0)
Adenoid tissue sequenced		
Yes, *n* (%)	40 (100)	-
Adenoid swab sequenced		
Yes, *n* (%)	14 (35)	-
No, *n* (%)	26 (65)	-
Sinus biopsy sequenced		
Yes, *n* (%)	-	5 (100)
Sinus wash sequenced		
Yes, *n* (%)	-	5 (100)
Atopy		
Asthma		
Yes, *n* (%)	1 (2.5)	0 (0)
No, *n* (%)	39 (97.5)	5 (100)
Allergic rhinitis		
Yes, *n* (%)	14 (35)	2 (40)
No, *n* (%)	26 (65)	3 (60)
Serum IgE test		
Yes, *n* (%)	7 (17.5)	5 (100)
No, *n* (%)	33 (82.5)	0 (0)
Mean ± SD (IU/mL)	258.8 ± 496.3	605.2 ± 903.8
Serum IgG test		
Yes, *n* (%)	2 (5)	4 (80)
No, *n* (%)	38 (95)	1 (20)
Mean ± SD (mg/dL)	647.5 ± 304.8	1178.3 ± 208.7
Serum IgA test		
Yes, *n* (%)	2 (5)	4 (80)
No, *n* (%)	38 (95)	1 (20)
Mean ± SD (mg/dL)	60.5 ± 60.1	148 ± 66
Serum IgM test		
Yes, *n* (%)	2 (5)	4 (80)
No, *n* (%)	38 (95)	1 (20)
Mean ± SD (mg/dL)	75.5 ± 10.6	173 ± 55
Medications		
Antibiotic		
Yes, *n* (%)	13 (32.5)	3 (60)
No, *n* (%)	27 (67.5)	2 (40)
Antihistamine		
Yes, *n* (%)	15 (37.5)	1 (20)
No, *n* (%)	25 (62.5)	4 (80)
Inhaled steroid		
Yes, *n* (%)	4 (10)	0 (0)
No, *n* (%)	36 (90)	5 (100)
Inhaled β2 antagonist		
Yes, *n* (%)	4 (10)	0 (0)
No, *n* (%)	36 (90)	5 (100)
Nasal steroid		
Yes, *n* (%)	6 (15)	1 (20)
No, *n* (%)	34 (85)	4 (80)
Leukotriene receptor antagonist		
Yes, *n* (%)	2 (5)	0 (0)
No, *n* (%)	38 (95)	5 (100)
Proton pump inhibitor		
Yes, *n* (%)	0 (0)	0 (0)
No, *n* (%)	40 (100)	5 (100)

**Table 2 microorganisms-11-00422-t002:** Microbial species identified from patients undergoing adenoidectomy and functional endoscopic sinus surgery (FESS) in the Johns Hopkins All Children’s Hospital (JHACH) cohort.

Species, *n* (%)	Adenoidectomy Patients *n* = 15	FESS Patients *n* = 5
*Streptococcus pneumoniae*	6 (40.0)	1 (20)
*Moraxella catarrhalis*	6 (40.0)	0 (0.0)
*Haemophilus influenza*	5 (33.3)	0 (0.0)
*Staphylococcus aureus*	2 (13.3)	0 (0.0)
*Streptococcus pyogenes*	1 (6.7)	0 (0.0)
*Pseudomonas aeruginosa*	1 (6.7)	0 (0.0)
Gram-positive cocci	1 (6.7)	0 (0.0)
*Curvularia* species	0 (0.0)	1 (20)

## Data Availability

Raw reads and detailed metadata of each sample from this study are available at NCBI’s Sequence Read Archive and BioSample database under the BioProject accession PRJNA869313.

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
