# Peer review of "Associations of Microbial Diversity with Age and Other Clinical Variables among Pediatric Chronic Rhinosinusitis (CRS) Patients"

_microorganisms, 2023, doi:10.3390/microorganisms11020422_

Round 1

Reviewer 1 Report

These 3 manuscripts would add to your references.

Pediatric Chronic Rhinosinusitis: Unmet Needs

by 

Russell J. Hopp

Department of Pediatrics, Division of Allergy, Pulmonology and Sleep Medicine, University of Nebraska Medical Center and Children’s Hospital and Medical Center, Omaha, NE 68198, USA

Sinusitis 20204(1), 2-7; https://doi.org/10.3390/sinusitis4010002

Received: 29 April 2020 / Revised: 15 May 2020 / Accepted: 8 June 2020 / Published: 10 June 202

Fifty Years of Chronic Rhinosinusitis in Children: The Accepted, the Unknown, and Thoughts for the Future

Russell J. Hopp  Jenna Allison , and David Brooks

Published Online:https://doi.org/10.1089/ped.2016.0645

Do Adult Forms of Chronic Rhinosinusitis Exist in Children and Adolescents?

by 

Russell J. Hopp

Department of Pediatrics, Creighton University School of Medicine, 2500 California Plaza, Omaha, NE 68131, USA

Sinusitis 20172(4), 7; https://doi.org/10.3390/sinusitis2040007

 1.  The recruitment of subjects section was IMPOSSIBLE to follow.  Must be clarified.  Seems like it's a study within a study.... and then UPMC was added

2.  is it necessary to bring the UPMC data into this study????  made it too confusing.  Can add as a Discussion item but not data comparison.

3.  Explain alpha and beta diversity somewhere!

3a.  Your groupings are too small (ICS use group)

4,  Was alpha diversity with the JHACH group or between the UPMC and JHACH groups?  Figure 1..... although the values on the y axes is not very familiar to most readers.

5.  Same ? on beta diversity

6.  All the Figures drag the UPMC data set into the Results.  MUST explain why and only Figure 5 and 6 was of much value.

7.  I think you brought UPMC raw data into the study and then bounced back and forth.  Not happy about this.  If your data results support their data, fine, but if you data set is too low, (and you say that in the conclusion) just adding together data is suspect.

Figure 5 and 6 support the comparison.  Lumping their raw data and your raw data into Figures 1-4 is my issue.  Your  data can support but re-analyzing UPMC data or cross-comparing is my issue as a Reviewer.

Reviewer 2 Report

Peer review of Shen Jean Lim et al, Associations of microbial diversity with age and other clinical variables among pediatric chronic rhinosinusitis (CRS) patients. This manuscript analyzed the bacterial diversity and genera of adenoid tissue, adenoid swab, maxillary sinus, and sinus wash samples from 45 pediatric CRS patients versus healthy controls. I would suggest the author make changes in the description and reconfirm the text/results to improve the manuscript.

My major concerns:

1.    In section 2.1, I would suggest the author modify or clarify the description of “9 months to 55 years (inclusion criteria)” as participants were all derived from pediatrics.

2.    It’s hard to understand the design of this study. I would suggest the author clarify the proportion of different specimens (adenoid tissue, adenoid swab, maxillary sinus, and sinus wash) in each group by using table (table 1) or figure. In addition, the healthy controls were not recruited from this study. If possible, the author should also describe the demographics of the study population in the UPMC cohort by using the table (JHACH cohort versus UPMC cohort).

3.    The author should discuss/explain the results of higher alpha diversity with pre-operative nasal steroid use in the section of the discussion.

4.    What is the clinical implications/meaning of the association between microbial diversity/microbiome and age in pediatric CRS patients (should be described in the section of discussion). Most importantly, what is the clinical implications/meaning of figure 4? There was no difference between CRS and healthy under the concept of correlations between microbiome and age.

5.  According to figures 5 and 6, the author should be more specific display the results (by using a table or figure): (1) in contrast to the UPMC control, which microbiome is critical to JHACH CRS and also be proved in UPMC CRS; (2) in contrast to the UPMC control and UPMC CRS, which microbiome is unique to JHACH CRS.

6.    In the description of “Streptococcus, Moraxella, and Haemophilus were the most abundant microbiome in the JHACH cohort”; however, (1) there were only 15 participants with the results of sinus culture; (2) it conflicts with the description in line 419: the lower Moraxella abundances (extension to the question 5). How about the results between participants with and without antibiotic/nasal steroid use?

Round 2

Reviewer 1 Report

Reference # 2 is not complete.

Author Response

We fixed reference #2.

Reviewer 2 Report

The author has responded all my concerns well.

Author Response

We thank the reviewer for the favorable review.